# Simulating Fiber-Reinforced Concrete Mechanical Performance Using CT-Based Fiber Orientation Data

**DOI:** 10.3390/ma12050717

**Published:** 2019-03-01

**Authors:** Vladimir Buljak, Tyler Oesch, Giovanni Bruno

**Affiliations:** 1Faculty of Mechanical Engineering, University of Belgrade, Kraljice Marije 16, 11120 Belgrade 35, Serbia; 2Bundesanstalt für Materialforschung und–prüfung, BAM (Federal Institute for Materials Research and Testing), 12205 Berlin, Germany; Tyler.Oesch@bam.de (T.O.); Giovanni.Bruno@bam.de (G.B.); 3Institute of Physics and Astronomy, University of Potsdam, Karl-Liebknecht-Str.24-25, 14476 Potsdam, Germany

**Keywords:** Fiber-reinforced concrete, X-ray computed tomography (CT), anisotropic fiber orientation, inverse analysis

## Abstract

The main hindrance to realistic models of fiber-reinforced concrete (FRC) is the local materials property variation, which does not yet reliably allow simulations at the structural level. The idea presented in this paper makes use of an existing constitutive model, but resolves the problem of localized material variation through X-ray computed tomography (CT)-based pre-processing. First, a three-point bending test of a notched beam is considered, where pre-test fiber orientations are measured using CT. A numerical model is then built with the zone subjected to progressive damage, modeled using an orthotropic damage model. To each of the finite elements within this zone, a local coordinate system is assigned, with its longitudinal direction defined by local fiber orientations. Second, the parameters of the constitutive damage model are determined through inverse analysis using load-displacement data obtained from the test. These parameters are considered to clearly explain the material behavior for any arbitrary external action and fiber orientation, for the same geometrical properties and volumetric ratio of fibers. Third, the effectiveness of the resulting model is demonstrated using a second, “control” experiment. The results of the “control” experiment analyzed in this research compare well with the model results. The ultimate strength was predicted with an error of about 6%, while the work-of-load was predicted within 4%. It demonstrates the potential of this method for accurately predicting the mechanical performance of FRC components.

## 1. Introduction

With recent developments in structural and material mechanics, assessments of safety margin with respect to non-linear system response and failure, instead of admissible stresses, became possible and even required by several codes [1,2,3]. Numerical methods for the accurate simulation of the non-linear behavior of engineering structures have also been developed in last few decades and incorporated into computational tools [4]. This evolution has significantly increased the need for knowledge about the inelastic properties of materials (e.g., plasticity, damage, creep, fracture, etc.) which cannot be assessed, unlike the elastic parameters, by means of non-destructive tests such as those based on ultrasound tests [5]. Furthermore, the assessment of inelastic properties when combined phenomena take place (e.g., plasticity with damage and fracture) is rather difficult, or even impossible, using standardized tests for the evaluation of compressive or tensile strength as a single material property [6,7].

Accurate numerical modeling within the non-linear regime is related to the appropriate selection of a constitutive model, capable of accounting for phenomena that are taking place at the material level (e.g., plastic deformation, damage of the material, creep, etc.). Such a constitutive model would offer a framework for the accurate modeling of a structural response in the general context, beyond the one represented by the experiment performed for its calibration. Therefore, the quantification of the parameters that govern the constitutive equations should not be merely reduced to the fitting of a single experimental response.

The importance of appropriate constitutive model selection becomes more evident when a complex material, such as fiber-reinforced concrete (FRC), should be modeled. Owing to the presence of small fibers, the structural response of FRC with respect to conventional reinforced concrete is considerably different. With conventional reinforcement, significant elongation of the steel bar is required, so that it can carry tensile loads, which requires the notable opening of macro cracks within the concrete. In contrast, in FRC the cracks are often barely visible to the naked eye, and are developed in the form of a network, which gives the structural member greater ductility and, at the same time, limits the exposure of fibers to the ambient conditions [8,9].

In previous years, considerable research efforts have been devoted to studying the mechanical response of FRC. Significant attention has been devoted to analyzing the influence of fiber orientation on the mechanical response of structures [10,11]. Since it is recognized that fiber distribution and orientation play an important role in global mechanical properties, several authors have discussed the influence of the casting process on the orientation of fibers [12], analyzed various methods to measure it [13], and tried to predict it through flow simulations [14]. There have also been many experimental studies focused on the quantification of the global mechanical properties of structural components made of FRC [15,16,17]. However, for the systematic incorporation of this material into structural analysis, a proper constitutive description and related parameter calibration is required. 

The mechanical response of the structural components made of FRC depends, to a large extent, on the local distribution and orientation of reinforcing fibers. Such information can be collected through the use of X-ray computer tomography (CT), but its effective incorporation into numerical modeling still needs to be solved. The major difficulty for successful modeling is related to the fact that the existing orthotropic constitutive damage models, which are implemented in commercial finite element codes, are suitable for defining anisotropic material behavior only at the structural level. While this can be an appropriate strategy to model conventional reinforced concrete, it is not appropriate for the FRC, where locally strengthened directions, achieved by reinforcing fibers, vary considerably within different regions of individual structural components.

For reliable numerical simulations, fiber distribution and orientation should be included within the constitutive description, thus requiring multi-scale approaches with the capability of incorporating the inherent variability of the internal structure. For this purpose, discrete models [18,19], can be used, with further modifications, to take into account fiber distribution and orientations. These discrete models are capable of addressing material behavior at the micro and meso-scales. For the macro-scale, however, which is of importance for the analysis of large-scale structures, it is desirable to have a continuum phenomenological model. Such models are based on a representative volume element (RVE), treated as a continuum, without the necessity to model smaller constituents (e.g., fibers or grains) [20]. The presence of these individual constituents is, instead, taken into account through homogenized, macro-scale mechanical characteristics. These models, necessarily, involve certain assumptions that could limit their applicability. The feasibility of the numerical implementation, however, is significantly improved, since the problem is solved on a single scale. This approach is adopted in the present study. 

Considering the nature of the phenomena that take place on the fiber scale, a reasonable approach would be to employ a damage model. The major difficulty related to the employment of existing constitutive damage models within commercial finite element modeling (FEM) codes is that even though they can simulate either isotropic or orthotropic behavior, the orthotropic behavior can only be modeled along the directions defined at the structural level. This can be an appropriate strategy to model conventional reinforced concrete, where reinforcing bars have well established directions with respect to the structure, but is not sufficient for FRC, where locally strengthened directions change from one point to another.

A detailed description of the FRC material as well as the characteristics of the CT measurements and the analysis of fiber orientation data are presented in Section 2. Section 3 is devoted to a description of the main features of the finite element model used in this study and its specialized adaptation to take into account the local material variability with reference to the three-point bending experiment of a notched beam, which was adopted as the test for subsequent material calibration. The inverse analysis procedure developed to assess governing constitutive parameters is outlined in Section 4. Section 4 presents the results produced by the calibrated constitutive model including those from its validation. Section 5 concludes the paper by offering a brief summary of the advantages and limitations of the proposed strategy together with some future prospects. 

## 2. Materials, Computed Tomography Measurements and Data Reconstruction 

### 2.1. Material and Specimen Properties

The FRC material used during this research program is an ultra-high-performance concrete (UHPC) developed by the US Army Engineer Research and Development Center (ERDC, Vicksburg, MS, USA). As a reactive powder concrete mixture, this material contains no course aggregate particles and is characterized by a low water-to-cement ratio and high cement paste content [21,22,23]. This material has a nominal compressive strength of 200 MPa and contains nominally 3.6% steel fibers by volume [24]. The steel fibers were 30 mm long, 0.55 mm in diameter, and had hook-like pre-deformations at each end.

The two beams analyzed in this study were nominally 220 mm long, 48 mm high, and 30 mm wide. After curing, nominally 18 mm deep by 5 mm wide “notches” were saw cut at the bottom of each beam midway along the length. Given the large aspect-ratio of the beams and the relatively small height and width dimensions relative to fiber length, it was expected that significant anisotropies in fiber orientation could occur, in particular with the fibers oriented principally in the direction of the beam length. Following CT scanning, the specimens were loaded to failure in a three-point bending configuration. Further detail about the materials, specimens, and mechanical testing was reported by Williams, Roth, Trainor, et al. [22,23,24,25]. 

### 2.2. CT Measurement and Analysis

Prior to mechanical testing, the central section of each beam was scanned using CT; for details of the specific scanning procedure and parameters [24]. Three-dimensional reconstruction of the CT images was completed using a proprietary algorithm provided by the CT instrument vendor (Northstar Imaging, Inc., Rogers, MN, USA) 

Analysis of fiber orientation, including the compilation of fiber data into a finite element mesh, was completed using the Fiber Composite Material Analysis module of VGSTUDIO MAX [26]. A section of the CT image from a beam with overlaid finite element mesh can be seen in Figure 1.

### 2.3. Tensor-Based Analysis of Fiber Orientation

In order to achieve a compact description of fiber orientation, a tensorial representation was adopted. Specifically, for a unit vector $p=[p_{1},p_{2},p_{3}]$, the second-order orientation tensor was constructed as its dyadic product [27] namely:(1)$$T=\left[\begin{array}{ccc} a_{11} & a_{12} & a_{13}\\ a_{21} & a_{22} & a_{23}\\ a_{31} & a_{32} & a_{33} \end{array}\right]=\left[\begin{array}{ccc} p_{1}^{2} & p_{1}p_{2} & p_{1}p_{3}\\ p_{2}p_{1} & p_{2}^{2} & p_{2}p_{3}\\ p_{3}p_{1} & p_{3}p_{2} & p_{3}^{2} \end{array}\right]$$

The eigenvalue analysis of matrix **T** recovers the unit vector **p**, which is the eigenvector corresponding to the largest eigenvalue. In this case, this is the only eigenvalue larger than zero (and equal to 1). For a group of N fibers, each is associated with a fiber unit vector **p**^i^, i=1, …N describing its orientation. The average orientation tensor is computed by taking the mean value of entries from the individual orientation tensors, calculated for each **p**^i^ [27], namely:(2)$$T^{avg}=\left[\begin{array}{ccc} a_{11}^{avg} & a_{12}^{avg} & a_{13}^{avg}\\ a_{21}^{avg} & a_{22}^{avg} & a_{23}^{avg}\\ a_{31}^{avg} & a_{32}^{avg} & a_{33}^{avg} \end{array}\right]\;,\;\mathrm{with}\;a_{ij}^{avg}=\frac{1}{N}\sum_{k=1}^{N}a_{ij}^{k}$$

By performing the eigenvalue analysis of $T^{avg}$, a set of three eigenvectors and corresponding eigenvalues is obtained. For an arbitrary case, when fibers from the group are not all parallel, all the three eigenvalues are different from zero, while their summation is equal to one. The magnitude of each eigenvalue gives the statistical proportion of the fibers from the analyzed group, aligned along the corresponding eigenvector. Thus, by averaging the orientation of the analyzed fiber set, the smallest error is introduced if the representative direction is taken to be the eigenvector corresponding to the largest eigenvalue. This conclusion stems directly from proper orthogonal decomposition (POD) theory [28,29]. Within POD, it is demonstrated that when representing a set of vectors $p^{i}={\left[p_{1}^{i},p_{2}^{i},p_{3}^{i}\right]}^{T}$ with one component approximation, the error of approximation is minimized if they are projected to the direction of the eigenvector corresponding to the largest eigenvalue of matrix **D**: (3)$$D=\left[\begin{array}{ccc} d_{11} & d_{12} & d_{13}\\ d_{21} & d_{22} & d_{23}\\ d_{31} & d_{32} & d_{33} \end{array}\right]=\left[\begin{array}{cccc} p_{1}^{1} & p_{1}^{2} & \ldots & p_{1}^{N}\\ p_{2}^{1} & p_{2}^{2} & \ldots & p_{2}^{N}\\ p_{3}^{1} & p_{3}^{2} & \ldots & p_{3}^{N} \end{array}\right]\cdot \left[\begin{array}{ccc} p_{1}^{1} & p_{2}^{1} & p_{3}^{1}\\ p_{1}^{2} & p_{2}^{2} & p_{3}^{2}\\ \ldots & \ldots & \ldots \\ p_{1}^{N} & p_{2}^{N} & p_{3}^{N} \end{array}\right]$$

It is easily demonstrated that matrices **D** and $T^{avg}$ are related through a scaling factor 1/N, since $a_{ij}^{avg}=\frac{1}{N}\sum_{i=1}^{N}p_{i}^{k}p_{j}^{k}$, while $d_{ij}^{avg}=\sum_{i=1}^{N}p_{i}^{k}p_{j}^{k}$, and, hence, have the same eigenvectors. 

The developed numerical model uses two-dimensional domains. Volume cells, over which fiber orientation is analyzed, are of the size 1 mm × 1 mm in the modeled plane and have a 30 mm thickness. Fibers crossing this volume are averaged according to the above outlined procedure, and replaced by the first two components of the first eigenvector (i.e., projected to modeling plane x-y), see Figure 2. 

The averaging of fiber properties over the 30 mm cell thickness, corresponding to the width of the entire beam, is the required step for implementing the real, 3D fiber measurements into the 2D numerical model.

## 3. Modeling

### 3.1. Constitutive Model for Progressive Damage of Fiber-Reinforced Concrete

In this paper, a fully phenomenological orthotropic damage model is adopted to model FRC (i.e., a model that accounts for the effect of fibers and not their physical presence). The presence of fibers contributes to the increase in load-carrying capacity of the whole specimen. This increase is manifested only along the direction of fibers, while in the perpendicular directions the response is considered to roughly correspond to that of unreinforced concrete (though possibly somewhat weaker). Therefore, to model the response employing the phenomenological constitutive model, it is appropriate to use the orthotropic mode: a model with different behavior along different, well-defined directions. The local variability of the structure, in terms of fiber orientation, is incorporated by previous pre-processing. This is achieved by dividing the portion of the sample most susceptible to progressive damage, due to formation of a crack network, into 1 mm × 1 mm 2D elements. Within each of these regions, local coordinate systems are assigned based on fiber-orientation data collected from CT measurements, averaged over the considered region. Thus, through the use of local coordinate systems, the primary direction strengthened by the reinforcement of the fibers in each individual zone is simulated. The constitutive parameters are the same for the whole sample, while local structural variability is accounted for through specific coordinate systems. This formulation results in a constitutive model capable of predicting unique global specimen behavior, since the variability at the structural level, including significant differences in overall structural response, result from changes in these local orientations. Such an approach provides the advantage that the problem is solved over one scale only, requiring the quantification of only one set of constitutive parameters, here solved on the basis of the designated inverse analysis procedure. 

The adopted constitutive model is typical for fiber-reinforced composites, existing in the commercial FEM code ABAQUS (Dassault systemes, Providence, RI, USA) [4]. These materials usually exhibit elastic-brittle behavior, with the damage initiation without any previous plastic deformation. Damage here refers to the onset of degradation at a material point, implemented within the continuum constitutive model through the reduction of elastic constants. Given the significant difference in the reinforcing behavior along the axis of the fiber compared to the reinforcing behavior in the direction perpendicular to this axis, an orthotropic damage model is adopted. Within this model, stresses are related to the total strains through following relation:(4)$$\sigma =C_{D}\cdot \varepsilon$$
with the elasticity matrix computed by:(5)$$C_{D}=\frac{1}{D}\left[\begin{array}{ccc} (1-d_{f})E_{1} & (1-d_{f})(1-d_{m})\nu_{21}E_{1} & 0\\ (1-d_{f})(1-d_{m})\nu_{12}E_{2} & (1-d_{f})E_{2} & 0\\ 0 & 0 & (1-d_{s})G\cdot D \end{array}\right]$$
where D is calculated as:(6)$$D=1-\left(1-d_{f}\right)\left(1-d_{m}\right)\nu_{12}\nu_{21}$$

The material mechanical response is therefore governed by the following parameters:*E_1_*—Young’s modulus for longitudinal direction (i.e., along axis of the fiber)*E_2_*—Young’s modulus for transversal direction (i.e., corresponding to unreinforced concrete properties)*ν_12_*, *ν_21_*—Poisson’s ratios *d_f_*, *d_m_* and *d_s_*—Damage variables for the longitudinal load capacity, the transversal load capacity and for shear capacity, respectively 

In the general version of the model, the two Young’s moduli are not the same, however within this study, the model is simplified by assuming the same value for both moduli. The damage parameters are bounded by the initial value of zero, corresponding to “virgin” material, and the maximum value of one, corresponding to fully degraded material. The calculation of each particular damage parameter value was related to the formulation of the damage initiation criterion (i.e., the state of stress at which the damage parameter value begins to increase above zero) and the criterion by which the damage parameter evolves up to a value of one, representing complete failure. This second damage evolution criterion completely governed the post-damage behavior of the material in the model.

The model adopted in this study (outlined above) required, besides the elastic parameters, the definition of the following parameters governing the damage: Damage initiation relative to the longitudinal tensile capacity (and, thus, an increase in *d_f_*) was assumed to occur once the longitudinal tensile stress exceeds the predefined value of σ_f_, corresponding to the beginning of fiber reinforcement failure. Here, fiber reinforcement failure is taken to mean the point at which the load carrying capacity of the fibers begins to decrease. This may be due to one or more phenomena (for instance excessive plastic deformation of the fibers, slip along the fiber-mortar interface, etc.);Damage initiation relative to the transverse tensile capacity (and, thus, an increase in *d_m_*) was assumed to occur once the transverse tensile stress exceeded the predefined value of σ_c_, corresponding to the beginning of unreinforced concrete tensile failure. In other words, fibers are assumed not to contribute to strengthening in the direction perpendicular to their long axis;Damage initiation relative to compression capacity is defined identically for both the longitudinal and transverse load capacities (corresponding to an increase in *d_f_* or *d_m_*, respectively). This initiation was assumed to occur once the compressive stress in a given direction exceeded the predefined value of *σ_cmp_*, corresponding to the beginning of unreinforced concrete compression failure. This simplified damage initiation criterion relies on the assumption that there is no significant contribution of fibers relative to compression failure; Damage initiation relative to the shear capacity (corresponding to an increase in *d_s_*) was assumed to occur once the shear stress exceeds the predefined value of *σ_SL_* or *σ_ST_*, corresponding to the beginning of concrete shear failure in the longitudinal or transversal directions, respectively.It was assumed that the damage propagation was to be governed by the fracture energy dissipated up to the full degradation of the element. In the general version of the model, separate fracture energy parameters governed the damage propagation in longitudinal (*d_f_*), transversal (*d_m_*) and shear (*d_s_*) loading. In this study, however, in view of the 2D (i.e., simplified) nature of the model, a simplification was adopted by setting the value of each of these separate fracture energy parameters equal to the value of a single parameter, *G_f_*. Within the 2D model, it is not possible to model the complex crack pattern that is developed within the real, 3D sample. To compensate for this limitation of the model, it is expected that the *G_f_* parameter will be overestimated. Moreover, since it was expected that the response of the beam was to be dominated by the damage along the longitudinal direction of the fibers, the values of the fracture energy parameters for transversal and shear loading were considered of secondary importance to overall response. The effects of restricting all fracture energy parameters to a single value on the overall beam response was, consequently, considered to be negligible.

### 3.2. Finite Element Model of Three-Point Bending Test

The geometry of the three-point bending specimen considered in this model was adopted [24], specifically a beam with 48 mm × 30 mm rectangular cross-section and 220 mm overall length, with 200 mm of span between the supports. At the mid-span, below the point of load application, a notch 5 mm wide and 18 mm in depth was introduced into the beam. In order to significantly reduce the necessary computation time, a two-dimensional numerical model was implemented. The beam was modeled as a 2D plane-stress problem, while the cylinders, through which the loading was applied, were considered rigid analytical curves in unilateral contact with the deformable specimen. Between the specimen and the cylinders, contact with friction was assumed, with a Coulomb friction coefficient equal to 0.15, taken as an a priori known quantity. During a three-point bending test, the contact between the specimen and the supports does not exhibit significant sliding. The friction, therefore, does not have a large influence on the response and, thus, an ad hoc value suggested by the literature was assumed [30]. The modeled beam was divided into three zones. In the central zone, which was subjected to the largest value of bending moment, the orthotropic damage model described in Section 3.1 was implemented. In the two outer zones, which were subjected to far lower bending moments, a linear-elastic model was implemented (see Figure 3). The central zone included all material within 10 mm of each side of the notch, thus having the overall width of 25 mm. Due to the load pattern and the specific geometry of the considered specimen, it was expected that the cracks would be formed mostly within this zone. The finite element mesh was generated using the automatic advancing front algorithm in ABAQUS [4], which resulted in a somewhat different shape of the mesh in the two outer zones. However, these two regions of the beam, which are within the elastic range, exhibit rather small deformations and, therefore, do not significantly affect the solution. The finite element mesh selected for the model was verified through a usual procedure by comparing the results of the simulations of models with different mesh densities. Here specifically, the adopted model is compared against one with a significantly denser mesh (having an overall number of degrees of freedom (DOF) about 2.5 times larger). The larger numerical model led to results less than 1% different from those achieved by the adopted model with the coarser mesh. The comparison is not included in the paper for the sake of brevity. In what follows, a brief outline of the adopted damage constitutive model and its specialized adaptation in the present context is given. 

In order to take the local orientation of fibers into account, the central zone of the beam specimen was divided into 1 mm × 1 mm elements, each of these having its own unique local coordinate system. The local coordinate system of these elements was assigned such that its longitudinal (i.e., stronger) axis corresponded to the primary axis of the average orientations of the fibers within the element calculated using CT data from the specimen (see Figure 3 and Section 2.2). For regions without fibers, unreinforced concrete properties were assigned, thus providing the possibility for the model to also partially take into account fiber density distribution. Clearly, this approximation represents a simplification in view of the two-dimensional nature of the model. Variations in material properties (including fiber orientations) over the beam thickness were not considered within the 2D model (i.e., the volume of each element is 1 mm × 1 mm × 30 mm). This strategy served to test the proposed approach prior to its implementation within a more realistic three-dimensional model.

To verify the capability of the proposed approach to model diverse structural responses governed by local variability of fiber distribution, the following numerical exercise was performed. Two different numerical models of three-point bending beams were generated. The fiber distributions were arbitrarily selected to produce a significantly different mechanical response. Such variations in fiber orientation and distribution are commonly seen in FRC and result directly from variation in the material flow during the casting process [11,31]. The numerical simulations lead to the results depicted in Figure 4 and Figure 5. From these figures, it may be observed that a significantly different structural response is obtained in terms of both the overall force-displacement curve and the cracking pattern. 

Figure 4a depicts the model with gradually changing fiber orientations; Figure 4b depicts the model with all fibers oriented perpendicular to the longitudinal axis of the beam. The dimensions of the whole beam are given in Figure 3.

Numerical model (b) assumed all the fibers to be perpendicular to the longitudinal axis of the beam, therefore providing negligible reinforcing capacity for the three-point-bending load scenario. Indeed, as shown in Figure 5, the maximum force is significantly smaller than for the model (a). Furthermore, such uniform distribution does not provide any variability and, thus, the beam failed with one major crack in the mid-span (see Figure 4b). On the other hand, model (a) assumed that the fibers gradually changed orientation moving from left to right and from the bottom to the top of the sample, with certain zones not containing any fibers. Such a fiber distribution clearly leads to the strengthening of the specimen with respect to the model (b). Additionally, this fiber distribution produced a variation of the locally strengthened directions over the specimen. The effects of this characteristic were captured by the model and can be identified as a network of small, distributed cracks (depicted by green color in Figure 4a). 

## 4. Model Calibration and Validation

In the procedure implemented for this study, a three-point bending test, from which a force-displacement curve is obtained, is treated as the main experimental data. The measured fiber orientation characteristics for the beam used in this calibration are also, naturally, implemented into the numerical model and, thus, directly influence the corresponding constitutive parameter values. A discrepancy function is further constructed to quantify the difference between measured quantities and their computed counterparts [29]. Through the execution of a test simulation employing the above constitutive model, this function becomes dependent on the governing constitutive parameters. At this point, the discrepancy function is minimized with respect to the sought constitutive parameters. This is the solution of the inverse problem. 

Afterwards, the assessed parameter values can be treated as material representative data and can be used for arbitrary loading scenarios. Validation of the accuracy and resilience of the model was completed using CT-based fiber orientation measurements from a second three-point bending test in combination with the assessed material parameters to predict the structural response. This is the validation step [32].

### 4.1. Inverse Analysis Procedure for Quantification of Material Parameters

The reliability of numerical simulations of FRC structural components using the previously outlined damage model rests on the accuracy of the implemented constitutive parameters. In the present case, these parameters quantify the elastic response and progressive damage. In this context, further simplifications were adopted to reduce the number of necessary parameters: The values of Young’s modulus *E_1_* (fiber direction) and *E_2_* (transversal direction) were defined within the model based on experimental data, while the remaining elastic constants were considered to be a priori known values and were, correspondingly, fixed within the model. The response of the beam is dominantly governed by the damage parameters, regardless of the initial, perfectly elastic conditions. Thus, in order to reduce the number of parameters that had to be assessed, only one Young’s modulus value (identical for both *E_1_* and *E_2_*) was used. This simplification is reasonable, considering that the variation in Young’s modulus due to the presence of fibers has been previously reported as only about 10% of the Young’s modulus typical for unreinforced concrete [33].number of damage-initiation parameters were also defined in the model based on experimental data: the damage initiation stress in tension for the longitudinal direction (*σ_f_*), the damage initiation stress in tension for the transversal direction (*σ_c_*), the damage initiation stress in shearing for the longitudinal (*σ_SL_*) and for the transversal (*σ_ST_*) directions. The stress value for damage initiation in compression was assumed to be the same for both the longitudinal and the transversal directions and did not have to be directly calculated from the experimental data. This damage initiation stress in compression was therefore not subjected to the identification from the experiment and was taken to be eight times the magnitude of the defined damage initiation stresses for tension in the transversal direction, based on standard tensile-to-compression strength ratios described in ACI 318 [34]. This represents a reasonable simplification, assuming that there is no significant contribution of fibers to cracking loads in compression.

Using this formulation, the number of parameters that had to be defined based on experimental data was reduced to six: one elastic parameter, four damage-initiation parameters, and the fracture energy, which was described in Section 3.1. These parameters were quantified through an inverse analysis procedure completed using force-displacement data collected from the three-point-bending experiment. The main features of this inverse analysis procedure are briefly outlined in what follows. 

The unknown material parameters were calculated through an inverse analysis procedure in which a suitably selected discrepancy function designed to quantify the difference between measured and numerically computed quantities is minimized. The function has the following form:(7)$$\omega \left(p\right)={\left[u_{e}-u_{c}(p)\right]}^{T}\left[u_{e}-u_{c}(p)\right]$$

In this equation, p represented a variable vector, here specifically containing the six unknown material parameter variables. The vector u_e_ contained the force values at each of N points along the force-displacement curve, corresponding to the equidistant displacements, measured during the experiments. The vector u_c_ contains predictions for the force values at each of the N points. These predicted values were generated through test simulations, attributing to the material parameters estimated values corresponding to the current iteration of the optimization algorithm. In Figure 6, differences to be minimized are schematically visualized.

This function was numerically minimized by employing the “Trust Region” (TR) algorithm. Details of the TR algorithm and its numerical implementation are available in the literature [29,35,36], while in what follows, only the main features are outlined. 

The minimization started from an initial vector of the parameters. Within each iteration, a quadratic programming problem in two variable spaces was solved, namely a sub-space spanned by the gradient direction and the second derivative direction (called the Newton direction). The solution of the constrained minimization of this sub-problem provided the modified parameter vector for the next iteration (say **p**_k+1_) and resulted in an improved value of the discrepancy function (also called the objective function within the minimization community). Constraints are provided by a “trust region”, here adopted as a circle shape within the two-dimensional sub-space.

With such a formulation, the minimization problem was solved inside the trust region, in which the quadratic approximation was “trusted” to be a reasonably good approximation of the real discrepancy function. The quadratic approximation of ω(p) in the point **p**_k_ reached after the *k*^th^ iteration was generated by a means of the computed gradient in that point and by means of the Hessian matrix approximated through the Jacobean (**J**), namely:(8)$$\omega \left(p_{k}+\Delta p_{k}\right)\approx \omega \left(p_{k}\right)+\Delta p_{k}^{T}\cdot \frac{\partial \omega }{\partial p}\left(p^{k}\right)+\frac{1}{2}\Delta p_{k}^{T}\cdot H\cdot \Delta p_{k},\;\mathrm{where}\;H=J^{T}J$$

The described features of the TR algorithm clearly imply calculations of first derivatives only, numerically computed through finite differences, namely by separately varying each of the parameters. After individual simulations had been completed for variations in each separate parameter variable, a final simulation was completed with the revised values for all parameter variables applied. Therefore, the iterations contain *M*+1 simulations, where *M* is the number of parameters.

This iterative sequence was repeated until reaching the convergence criteria, here imposed as the change in the discrepancy function value and the difference between parameter vector norms between two consecutive iterations (specifically 1E-2, for the latter, and 1E-4, for the former, criterion). This was realized through an appropriate normalization of parameters with diverse orders of magnitude. The resulting parameter vector reached after a certain number of iterations represented the solution to the designed inverse problem and, therefore, also the representative material properties. As a remedy to the possible lack of convexity of the discrepancy function (7) and consequent termination in a local minimum, the TR iterative sequence was repeated starting from another initialization point. 

### 4.2. Calibration of the Model 

The inverse analysis procedure described in Section 4.1 was here employed to assess previously described unknown elastic and damage parameters of the outlined constitutive model. The experimental result used, as an input to the procedure, is a force-displacement curve collected from a three-point bending experiment on the same sample, described in Section 2.1. Based on comparisons of model predictions with this experimental data, a discrepancy function (7) was formed. Additionally, on the basis of CT measurements, the distribution of fiber orientation was calculated for the numerical model. The central zone of the beam, modeled by the damage constitutive model, was discretized by a finite element mesh with square elements 1 mm in size, resulting in the overall number of 1110 elements. To each of these elements, a unique local orientation was attributed in accordance with the procedure described in Section 2.3. This orientation corresponds to the fiber orientations measured in the specimen prior to testing. 

The inverse problem was found to be well-posed and the procedure converged to the solution after several iterations. The parameter values, which resulted as the solution to the inverse problem are listed in Table 1. The comparison between experimental and numerical force-displacement curves generated by employing these parameters is visualized in Figure 7. Good agreement between the two curves proves that the adopted approach is capable of capturing the overall mechanical response of three-point bending experiments. 

Considering the adopted simplifications outlined in Section 4.1, the assessed value of Young’s modulus represents the mean value of *E_1_* and *E_2_*. The resulting value of Young’s modulus was somewhat smaller than expected. This discrepancy is likely due to the influence of several factors on the displacement values measured during the experiments (e.g., compliance of the beam supports). The elastic properties, however, were not of primary interest here, since they can be measured more precisely with some alternative methods, therefore the removal of these effects was not considered. The damage initiation stresses are of the expected order of magnitude, when compared to the values reported in the literature [14]. The magnitude of parameter σ_f_ is comparable to flexural strengths previously measured for this material [23], while σ_c_ corresponds well to the tensile strength of unreinforced concrete. The identification procedure converged to these values without any preconditioning. A comment should also be made here regarding the parameter *G_f_*. The crack network developed in the two-dimensional model is significantly simpler than in the real three-dimensional case. Therefore, the only way for the numerical model to simulate the experimentally observed ductility is by increasing the value of *G_f_*. This could be confirmed by comparing the obtained values of *G_f_* with the absolute values of dissipated energy corresponding to different internal mechanisms [24]. 

### 4.3. Validation of the Model 

In order to verify the accuracy of the resulting material parameters for the given sample geometry and test setup, a second sample subjected to the three-point bending experiment is further considered. The two samples were made of nominally the same material, but with fairly different fiber orientation and distribution. Another numerical model was built, in which the fiber orientation was distributed within the model following the procedure described in Section 3.2 on the basis of CT measurements of the second beam sample before the test. The constitutive parameters were assumed to have the values resulting from the inverse analysis procedure described above, see Table 1. The experimental and numerical force-displacement curves for this second beam also turned out to be in quite good agreement (see Figure 8). This result corroborates the conclusion that the proposed strategy of modeling FRC by separating the material behavior (here expressed in terms of elastic-damage model) from the local structural variability, accounted for by applying different fiber orientations on an element-wise basis, is quite promising. The two considered specimens were different only in terms of fiber distribution, and two numerical models with the same material properties captured the unique structural response of each beam quite well. 

## 5. Closing Remarks and Future Prospects

This research showed that with a fully phenomenological model, without detailed modeling of fibers, the ductility of the fiber-reinforced concrete could be successfully simulated. Further, it is clear that such behavior can be captured with a single material property set, while the structural variability is incorporated through localized fiber orientation. The novelty of the proposed approach lies in the combination of the phenomenologically based model construction with the inverse analysis based calibration procedure. One obvious limitation of the modeling strategy presented here is the use of a simplified two-dimensional numerical model. The extension of this approach to a three-dimensional model requires the implementation of the proposed constitutive model using the USER subroutine of the commercial software ABAQUS. Such an extension would be a desirable development, but requires further research, which is presently ongoing, and additional numerical implementation that extends the relevant functionalities of the commercial software. The results achieved and presented within this paper, however, provide a meaningful contribution by demonstrating the advantages and limitations of this modeling approach using readily available numerical tools.

## Figures and Tables

**Figure 1 materials-12-00717-f001:**
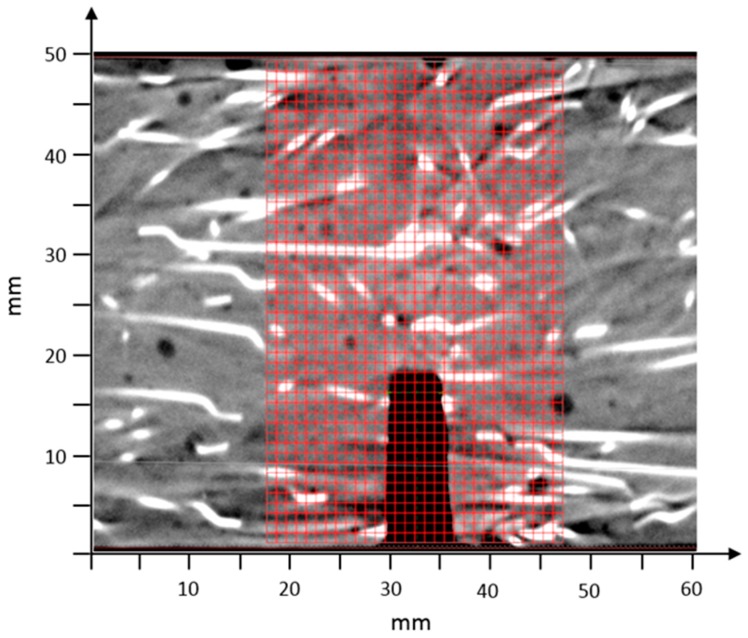
Section of beam computed tomography (CT) image with overlaid finite element mesh.

**Figure 2 materials-12-00717-f002:**
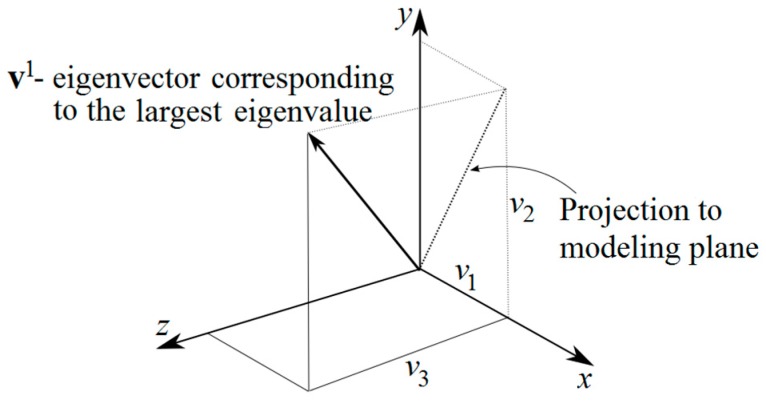
Eigenvector of principal fiber orientation.

**Figure 3 materials-12-00717-f003:**
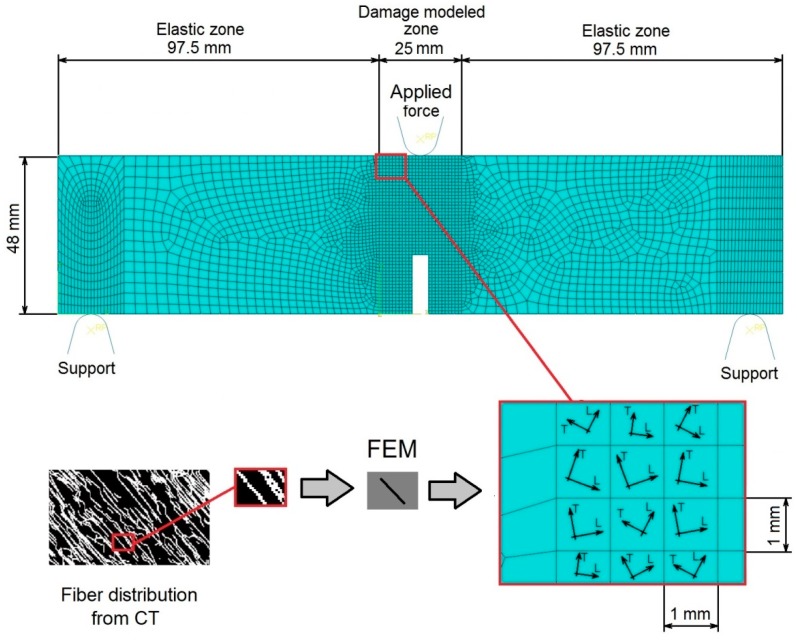
Adopted 2D finite element modeling (FEM) of three-point bending test with individual elements simulated by different constitutive models.

**Figure 4 materials-12-00717-f004:**
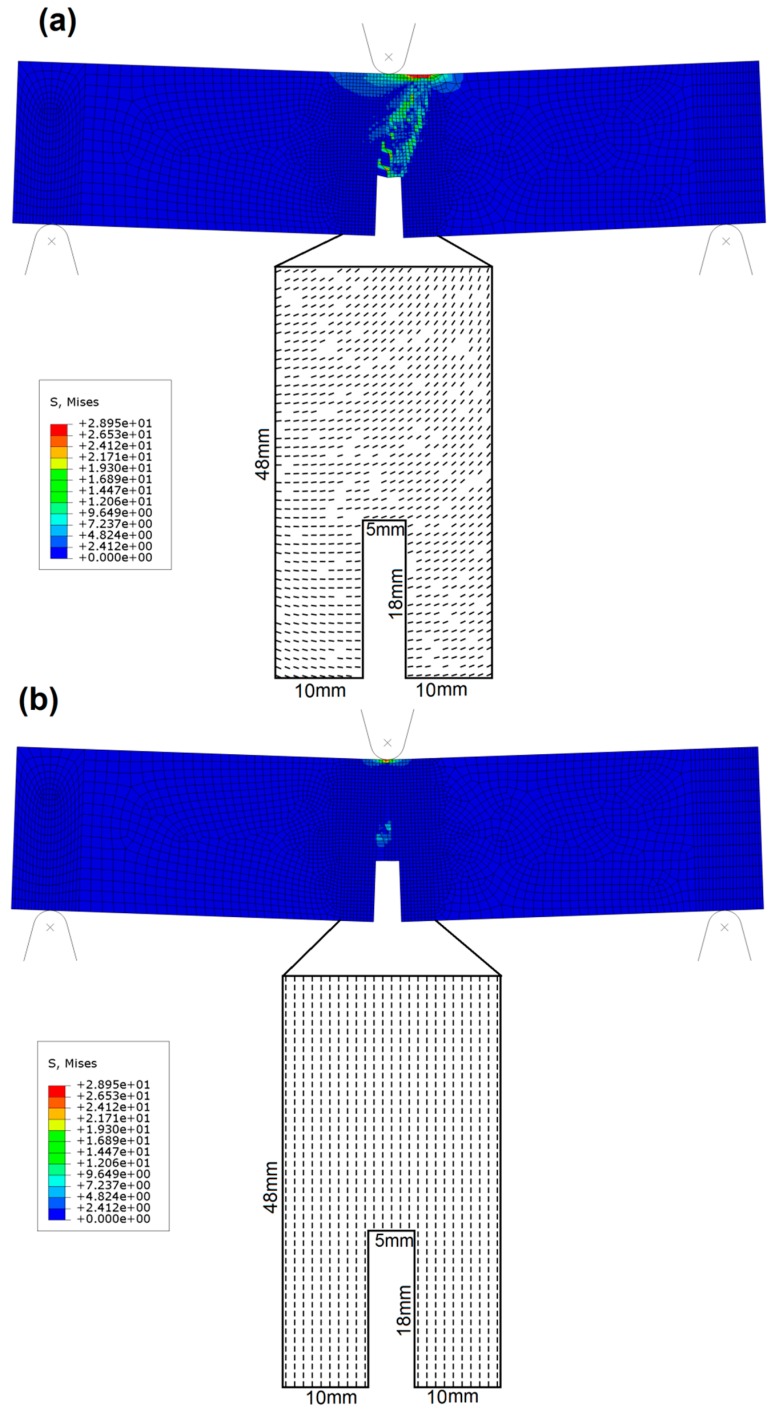
Two numerical models of the three-point bending test with different fiber orientations. (**a**) model with gradually changing fiber orientations, (**b**) model with all fibers oriented perpendicular to the longitudinal axis of the beam.

**Figure 5 materials-12-00717-f005:**
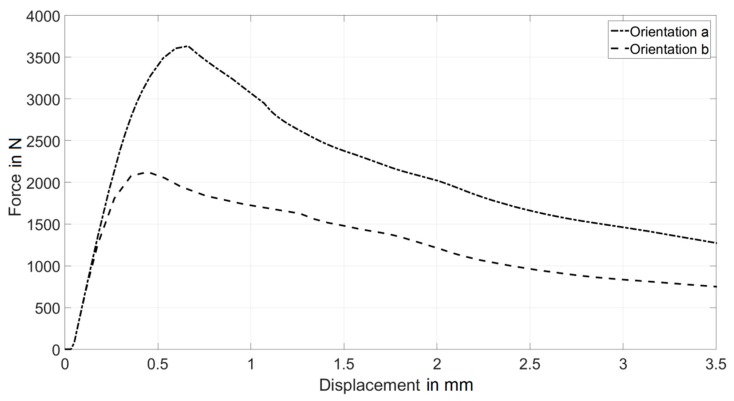
Force displacement curves resulting from the two numerical models of the three-point bending test with different fiber orientations.

**Figure 6 materials-12-00717-f006:**
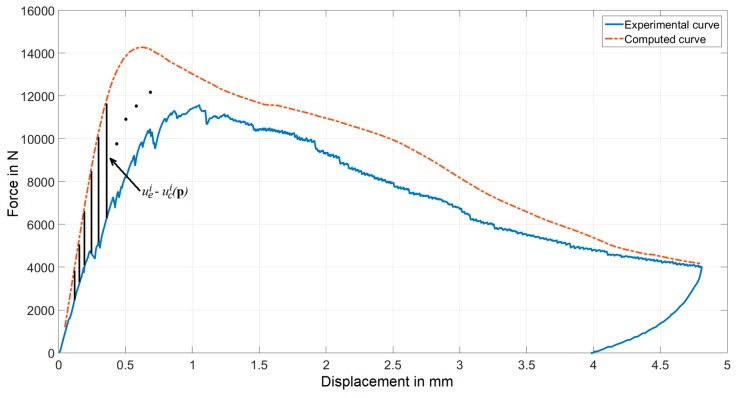
Construction of the discrepancy function from the force-displacement curve.

**Figure 7 materials-12-00717-f007:**
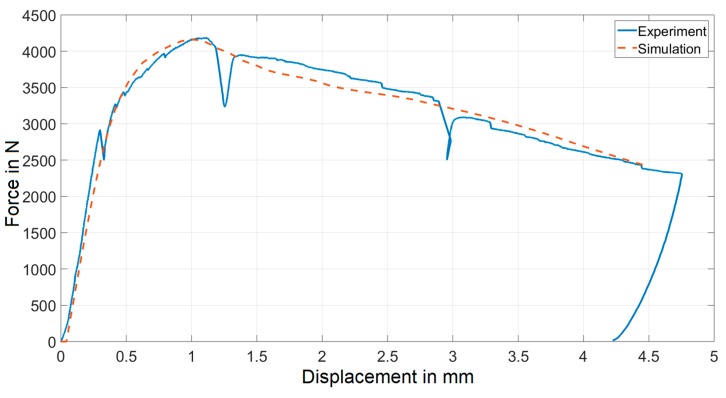
Comparison of force-displacement curves resulting from the experiment and resulting from simulations using inverse analysis.

**Figure 8 materials-12-00717-f008:**
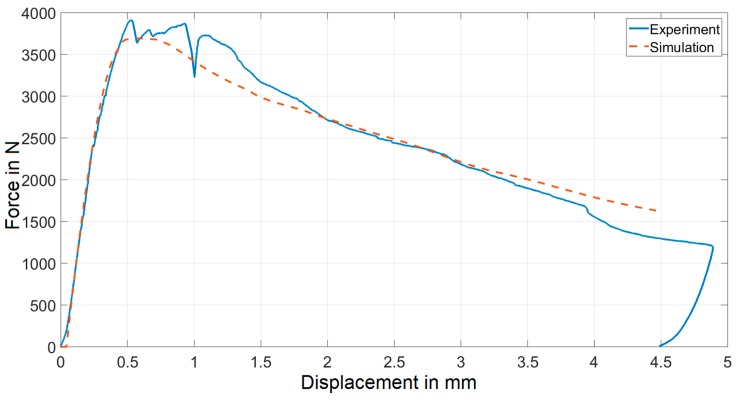
Comparison of experimental and simulated force-displacement curves for the second experiment.

**Table 1 materials-12-00717-t001:** Parameter estimates resulting from the inverse analysis.

Parameter	Resulting Value
*E_1/2_*	26.5 GPa
*σ* *_f_*	23.6 MPa
*σ* *_c_*	3.9 MPa
*σ* *_ST_*	2.7 MPa
*σ* *_SL_*	18.3 MPa
*G_f_*	27500 N/m

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
