# Peer review of "Simulating Fiber-Reinforced Concrete Mechanical Performance Using CT-Based Fiber Orientation Data"

_materials, 2019, doi:10.3390/ma12050717_

Round 1

Reviewer 1 Report

Very interesting paper with high scientific and language quality. There are just few suggestions for corrections and/or additional information.

1) line 140: Add a space between pi and [27]

2) line 234: It is described that a friction coefficient equal to 0,25 was used. Why did you use the coefficient equal to 0,15? Is it based on proposals in literature?

3) Figure 3: The FE-model in the elastic zone on left and right side of the beam is quite different. Why you did not use the same FE-network?

4) line 263: Correct the missed reference source

5) Figure 4a: The fibre orientation in Figure 4a) is quite special one. A comment is needed wants the reason for the selection of this fibre orientation.

6) line 418: Delete the unnecessary line break

Author Response

Manuscript  ID: materials-436925

Simulating fiber-reinforced concrete mechanical performance using CT-based fiber orientation data

by Vladimir Buljak, Tyler Oesch and Giovanni Bruno

Dear Editor,

We thank you and the reviewers very much for the appreciation of the manuscript quality, and for the constructive and valuable remarks and suggestions to improve our manuscript. We reply to the reviewers’ comments in detail in the following.

We amended all required figures and we inserted several new references to circumstantiate our points.

We trust the manuscript has greatly improved.

On the behalf of all authors

Best regards

Vladimir Buljak

Reviewer 1:

Comments and Suggestions for Authors

Very interesting paper with high scientific and language quality. There are just few suggestions for corrections and/or additional information.

1) line 140: Add a space between pi and [27]

The space has now been added.

2) line 234: It is described that a friction coefficient equal to 0,25 was used. Why did you use the coefficient equal to 0,15? Is it based on proposals in literature?

The selection of the friction coefficient value is now clarified and supported by an additional reference put in the manuscript.

3) Figure 3: The FE-model in the elastic zone on left and right side of the beam is quite different. Why you did not use the same FE-network?

The finite element mesh selected for the model is verified through a usual procedure by comparing the results of simulations from models with different mesh densities. Here specifically, the adopted model is compared against one with a significantly denser mesh (having an overall number of degrees of freedom (DOF) about 2.5 times larger). The larger numerical model led to results less than 1% different from those achieved by the adopted model with the coarser mesh. The comparison is not included in the paper for the sake of brevity. The mesh was generated using the automatic advancing front algorithm in ABAQUS (description and reference added to manuscript), which resulted in a somewhat different shape of the mesh in the two different regions. However, the two parts of the beam, which are within the elastic range, exhibit rather small deformations and, therefore, do not significantly affect the solution.

The text has been amended and the above explanation inserted (in brief form).

4) line 263: Correct the missed reference source

All the cross-references through the paper have been corrected.

5) Figure 4a: The fibre orientation in Figure 4a) is quite special one. A comment is needed wants the reason for the selection of this fibre orientation.

Both fiber orientations presented in Figure 4 are hypothetical. The idea was to generate them to be different from each other to evidence the capability of the model to account for the different macroscopic structural response related to different fiber distributions. A corresponding statement about this has now been added to the manuscript. Note that both the (a) and (b) models used the same material parameters, leading to different structural response, evidenced in the graph.

6) line 418: Delete the unnecessary line break

The line break has been deleted.

Reviewer 2 Report

please see file attached

Author Response

Manuscript  ID: materials-436925

Simulating fiber-reinforced concrete mechanical performance using CT-based fiber orientation data

by Vladimir Buljak, Tyler Oesch and Giovanni Bruno

Dear Editor,

We thank you and the reviewers very much for the appreciation of the manuscript quality, and for the constructive and valuable remarks and suggestions to improve our manuscript. We reply to the reviewers’ comments in detail in the following.

We amended all required figures and we inserted several new references to circumstantiate our points.

We trust the manuscript has greatly improved.

On the behalf of all authors

Best regards

Vladimir Buljak

Reviewer 2:

In this work a novel approach to model the material behavior of fiber reinforced concrete is presented combining a phenomenological approach and microstructural observations based on computed tomography. The topic is of interest and framed with the purpose of the journal. Before consideration I would like to ask to further clarify the following points:

Abstract: The abstract lacks any quantitative description of the accuracy of the modeling results. What do you mean by “compare well”?

According to the Journal requirements, the abstract should be as short as possible. However, since the reviewer’s point is valid, we made a compromise between this requirement and the reviewer’s suggestion, adding a sentence to the abstract that quantifies the accuracy of the ultimate strength and work-of-load model predictions.

Line 115: Could you please add any details on (expected) fiber orientation resulting from manufacturing of the specimens?

Given the large aspect-ratio of the beams and the relatively small height and width dimensions relative to fiber length, it was expected that significant anisotropies in fiber orientation could occur, in particular with the fibers oriented principally in the direction of the beam length.

A corresponding statement has been added to the text of the manuscript.

Figure 1: Please add a scale bar to the figure.

A scale bar has been added to the figure.

Line 154: After having read this paragraph is not quite clear why volume cells were used although modelling is based on two-dimensional domains. Please explain.

In order to extract fiber-orientation information from the CT images, the real three-dimensional sample was considered. All the fibers present within the sample were used in the averaging process. Since the numerical model was constructed within the two-dimensional domain using square cells, the fiber orientation used in each cell resulted from averaging over the 30 mm width of the real 3D volume cell.  A clarifying sentence has been added to the manuscript.

Line 163: Could you please specify why an orthotropic damage model is appropriate.

The presence of fibers contributes to the increase of load-carrying capacity of the whole specimen. This increase is manifested only along the direction of fibers, while in the perpendicular directions the response is considered to roughly correspond to that of unreinforced concrete (although probably somewhat weaker). If the mechanical behavior should be modeled by a phenomenological model (i.e. a model that accounts the effect of fibers and not their physical presence), then it is appropriate to use the orthotropic model: a model with different behavior along different, well-defined, directions.

A corresponding statement has been added to the manuscript.

Line 166: What do you mean by small? How big are the regions (one element) compared to fiber size? Please describe more precisely at this point of the manuscript.

The regions (i.e. elements) were 1mm × 1mm large, while the fibers were 30 mm long and 0.55 mm wide. This sentence has been revised in the manuscript to be more specific.

Line 189: Within section 3.1 E1 is not equal to E2, however in section 4 E1 is assumed to be equal to E2. Please explain.

The assumption of two equal Young’s moduli values is a further simplification of the model, which does not have to be made in its general presentation. The parameter assessment performed through mathematical programming requires calculation of derivatives with respect to the parameters at each iteration, achieved through finite differences. Therefore, the number of simulations increases with the number of assessed parameters. Considering that the elastic properties were not within the main focus of this work, and that the difference between the two moduli is expected to be around 10% (value reported in the literature) this assumption seemed to be quite reasonable.

A clarifying statement has been added to the manuscript.

Line 199: Please delete the “,”

The coma has been deleted.

Line 205: What do you exactly mean by “...beginning of fiber reinforcement failure.” Do you mean fiber breakage or interface failure? Please explain more precisely.

The beginning of fiber reinforcement failure here means the entering into the softening branch of the load carrying capacity, which is followed by some ductility. At the fiber level this can be due to multiple phenomena (e.g. excessive plastic deformation of fibers, beginning of interface failure, fiber damage, etc.). Within the employed phenomenological model, the distinction between different fiber failure mechanisms was not considered – only the macroscopic effect.  

A corresponding statement has been added to the manuscript.

Line 222: Please check layout, it seems that there are multiple spaces between “In” and “the”.

There is only one line spacing between the two words. The strange appearance seems to result from the “justification” settings in Microsoft Word (governed by the MDPI template). Presumably this problem will by fixed during the typesetting- and proof-phase of the publication process.

Line 224: Could you please explain why this simplification is valid.

Ductility observed on the structural level for a real sample is connected to the complex crack pattern that is developed. Such a pattern clearly cannot be realistically replicated by a two-dimensional model. In order to simulate global structural ductility, the local constitutive parameters (among them the fracture energy parameter) have to compensate for this limitation of the model. Due to this limitation of the model, it is expected that the value of the fracture energy parameter is overestimated. A comment regarding this type of error exists in the manuscript. Due to of its 2D nature, the model is not expected to accurately assess such a purely 3D-related parameter. Moreover, since the response of the beam was expected to be dominated by damage along the longitudinal direction of the fibers, the values of the fracture energy parameters for transversal and shear loading were considered of secondary importance to the overall response. This justifies the significant simplification to the model by setting all of the fracture energy parameters to be equal. The effect of this simplification on the overall beam response within the model was, consequently, considered to be negligible.

A few sentences have been added to clarify this issue.

Line 227: Please change mm to mm2

The units are now correctly written.

Line 234: Please add a reference for the considered value of the friction coefficient.

During the three-point bending test, the contact between the specimen and the supports does not exhibit significant sliding. The friction, therefore, does not have a large influence on the response. This is verified through sensitivity analysis. Therefore an ad hoc value suggested by literature (see the reference now inserted in the paper) was assumed.

Figure.3: Please add scale bars to the individual subfigures. In addition, please explain why the mesh differs between the support on the left and support on the right.

The finite element mesh selected for the model is verified through a usual procedure by comparing the results of simulations of models with different mesh densities. Here specifically, the adopted model is compared against one with a significantly denser mesh (having an overall number of degrees of freedom (DOF) about 2.5 times larger). The larger numerical model led to results less than 1% different from those achieved by the adopted model with the coarser mesh. The comparison is not included in the paper for the sake of brevity. The mesh was generated using the automatic advancing front algorithm in ABAQUS (description and reference added to manuscript), which resulted in a somewhat different shape of the mesh in the two different regions. However, the two parts of the beam, which are within the elastic range, exhibit rather small deformations and, therefore, do not significantly affect the solution. Scale bars have also been added to the figures, aside from the subfigure regarding the CT fiber distribution, which is a generic depiction.

Line 263: I assume that something went wrong with a reference. Please check.

The figure references have been corrected.

Figure 4 Please add a scale bar and a legend to better understand what is exactly presented in the FEM results. Please add difference in fiber orientation in caption (a)…., b) ….).

The figure 4 is now corrected according to the suggestion of reviewer. An additional sentence was introduced in the text to clarify the difference between the two fiber orientations.

Figure 5 Please check and change labeling of axis. According to DIN 461 square brackets are not appropriate.

Labeling of axes is changed now.

Line 275: It is not clear where fiber orientation considered for model (a) comes from. Is this the orientation defined with the aid of by CT images? Please explain.

Both fiber orientations presented in Figure 4 are hypothetical. The idea was to generate them to be different from each other to evidence the capability of the model to account for the different macroscopic structural response related to different fiber distributions. A corresponding statement about this has been added to the manuscript. Note that both the (a) and (b) models used the same material parameters, leading to different structural response, evidenced in the graph.

Line 273/278: The two sentences starting with “indeed” in line 273 and 278, respectively, express exactly the same. Please delete repetition.

The repetition has been deleted, and the concepts consolidated.

Line 283: It is not quite clear which fiber orientation was considered to define the constitutive parameters by solving the inverse problem. Could you please more precise on this information.

A few sentences were added to the text to clarify the origin of the fiber orientation used for the calibration experiment and for the control experiment.

Line 292: I would recommend to either delete the comma after the word “second” or to add a second comma after the word “control”.

The sentence was confusing. We have changed it into:

“Validation of the accuracy and resilience of the model was completed using CT-based fiber orientation measurements from a second three-point bending test in combination with the assessed material parameters to predict the structural response.”

Line 300: If only one Young’s modulus is considered for the entire structure, how do the authors account for a local fibrous reinforcement of the structure. Please be more clear on this point. Is there a significant effect if different moduli are considered even the difference might only be approximately 10%?

The response of the beam is governed primarily by the damage parameters, not by elasticity parameters. Regardless of the initial condition of the material, very quickly damage begins to reduce its strength. In addition, the difference in elastic properties reported  in the literature (and cited in the manuscript) is of about 10%. Thus, we made the simplification of assuming equal elasticity in all directions, as it is not critical for capturing accurate structural response. A few sentences have been added to the manuscript to clarify this issue.

Figure 6 see comment to Figure 5

The figure is now changed according to the Reviewer’s suggestion.

Figure 7 see comment to Figure 5

The figure is now changed according to the Reviewer’s suggestion.

Figure 8 see comment to Figure 5

The figure is now changed according to the Reviewer’s suggestion.

Round 2

Reviewer 2 Report

The manuscript has been significantly improved and now warrants publication in Materials